# Incidence of Antibiotic Exposure for Suspected and Proven Neonatal Early-Onset Sepsis between 2019 and 2021: A Retrospective, Multicentre Study

**DOI:** 10.3390/antibiotics13060537

**Published:** 2024-06-10

**Authors:** Liesanne E. J. van Veen, Bo M. van der Weijden, Niek B. Achten, Lotte van der Lee, Jeroen Hol, Maaike C. van Rossem, Maarten Rijpert, Anna O. J. Oorthuys, Ron H. T. van Beek, Gerdien H. Dubbink-Verheij, René F. Kornelisse, Laura H. van der Meer-Kapelle, Karen Van Mechelen, Suzanne Broekhuizen, A. Carin M. Dassel, J. W. F. M. Corrie Jacobs, Paul W. T. van Rijssel, Gerdien A. Tramper-Stranders, Annemarie M. C. van Rossum, Frans B. Plötz

**Affiliations:** 1Department of Paediatrics, Franciscus Gasthuis en Vlietland, Kleiweg 500, 3045 PM Rotterdam, The Netherlands; 2Department of Paediatrics, Erasmus MC University Medical Center, Sophia Children’s Hospital, Wytemaweg 80, 3015 CN Rotterdam, The Netherlands; 3Department of Paediatrics, Tergooi MC, Laan van Tergooi 2, 1212 VG Hilversum, The Netherlands; 4Department of Paediatrics, Amsterdam UMC, Emma Children’s Hospital, Meibergdreef 9, 1105 AZ Amsterdam, The Netherlands; 5Department of Paediatrics, Noordwest Hospital, Wilhelminalaan 12, 1815 JD Alkmaar, The Netherlands; 6Department of Paediatrics, Rijnstate Hospital, Wagnerlaan 55, 6815 AD Arnhem, The Netherlands; 7Department of Paediatrics, Zaans Medical Centre, Kon. Julianaplein 58, 1502 DV Zaandam, The Netherlands; 8Department of Paediatrics, Dijklander Hospital, Maelsonstraat 3, 1624 NP Hoorn, The Netherlands; 9Department of Paediatrics, Amphia Hospital, Molengracht 21, 4818 CK Breda, The Netherlands; 10Department of Paediatrics, Groene Hart Hospital, Bleulandweg 10, 2803 HH Gouda, The Netherlands; 11Department of Neonatal and Paediatric Intensive Care, Division of Neonatology, Erasmus MC University Medical Center, Sophia Children’s Hospital, Wytemaweg 80, 3015 CN Rotterdam, The Netherlands; 12Department of Paediatrics, Reinier de Graaf Gasthuis, Reinier de Graafweg 5, 2625 AD Delft, The Netherlands; 13Department of Neonatology, Maastricht University Medical Center (MUMC+), MosaKids Children’s Hospital, 6229 HX Maastricht, The Netherlands; 14Department of Paediatrics, Wilhelmina Hospital Assen, Europaweg-Zuid 1, 9400 RA Assen, The Netherlands; 15Department of Paediatrics, Deventer Hospital, Nico Bolkesteinlaan 75, 7416 SE Deventer, The Netherlands; 16Department of Paediatrics, Jeroen Bosch Hospital, Henri Dunantstraat 1, 5223 GZ Hertogenbosch, The Netherlands; 17Department of Paediatrics, Maaziekenhuis Pantein, Dokter Kopstraat 1, 5835 DV Beugen, The Netherlands

**Keywords:** early-onset sepsis, antimicrobial stewardship, neonates, anti-bacterial agent, blood culture, incidence

## Abstract

Management of suspected early-onset sepsis (EOS) is undergoing continuous evolution aiming to limit antibiotic overtreatment, yet current data on the level of overtreatment are only available for a select number of countries. This study aimed to determine antibiotic initiation and continuation rates for suspected EOS, along with the incidence of culture-proven EOS in The Netherlands. In this retrospective study from 2019 to 2021, data were collected from 15 Dutch hospitals, comprising 13 regional hospitals equipped with Level I-II facilities and 2 academic hospitals equipped with Level IV facilities. Data included birth rates, number of neonates started on antibiotics for suspected EOS, number of neonates that continued treatment beyond 48 h and number of neonates with culture-proven EOS. Additionally, blood culture results were documented. Data were analysed both collectively and separately for regional and academic hospitals. A total of 103,492 live-born neonates were included. In 4755 neonates (4.6%, 95% CI 4.5–4.7), antibiotic therapy was started for suspected EOS, and in 2399 neonates (2.3%, 95% CI 2.2–2.4), antibiotic treatment was continued beyond 48 h. Incidence of culture-proven EOS was 1.1 cases per 1000 live births (0.11%, 95% CI 0.09–0.14). Overall, for each culture-proven EOS case, 40.6 neonates were started on antibiotics and in 21.7 neonates therapy was continued. Large variations in treatment rates were observed across all hospitals, with the number of neonates initiated and continued on antibiotics per culture-proven EOS case varying from 4 to 90 and from 4 to 56, respectively. The high number of antibiotic prescriptions compared to the EOS incidence and wide variety in clinical practice among hospitals in The Netherlands underscore both the need and potential for a novel approach to the management of neonates with suspected EOS.

## 1. Introduction

Antibiotics are the most commonly prescribed drugs in neonatal care units [1,2,3,4]. The clinical challenge of predicting which neonates will develop a life-threatening infection often leads to a cautious approach, with a tendency to prescribe antibiotics at a relatively low threshold. Conversely, the growing understanding of the association between early-life antibiotic usage, its adverse effects on the microbiome, and the subsequent development of diseases like asthma and diabetes later in life, underscores the importance of significantly reducing unnecessary antibiotic prescriptions [5,6,7,8,9,10,11,12,13,14,15].

It is suggested that to successfully improve antibiotic stewardship, the initial step is measurement and dissemination of data on one’s own current performance [16,17]. This applies as a starting point from where collaboration and benchmarking can take place and eventually novel algorithms can be developed to elevate decision making to a more evidence-based and factual approach [16]. Relatively recent data on antibiotic use and the incidence of EOS are available for a select group of countries, including high-income countries in Europe, Australia and North America. These data show large variations in the number of antibiotic prescriptions between different networks, which may be due to a variety of reasons, such as differences in health care systems and populations [18,19,20,21]. It is therefore important for countries to measure their own performance and share these data to understand international variation. 

However, there has been no recently published extensive assessment of antibiotic usage for suspected EOS in neonates born in The Netherlands. Therefore, this study aims to measure antibiotic initiation and continuation for suspected EOS, as well as EOS incidence, in The Netherlands. Additionally, it seeks to investigate variations in treatment practice across hospitals, anticipating differences due to low adherence to the current national guideline [22]. Furthermore, the study aims to understand the differences in EOS incidence and treatment between regional and academic hospitals, expecting higher EOS incidence and treatment rates in academic hospitals due to the generally younger and medically more complex population [23,24,25,26,27]. This study may serve as a model for other countries seeking to pragmatically identify the level of overtreatment as baseline for improvement. 

## 2. Results

Of the 70 hospitals that were approached for participation, 15 hospitals were included in the study (Figure 1). Among participating hospitals, there were 13 regional hospitals equipped with Level I-II facilities and 2 academic hospitals equipped with Level IV facilities. An overview of all participating hospitals, including their local agreements on minimal gestational age at admission, the province they are located and average birth rate per year can be found in Appendix A. 

### 2.1. Overall Initiation and Continuation of Antibiotic Therapy, and Proven EOS Incidence

Between 1 January 2019 and 31 December 2021, a total of 103,492 neonates were born across all participating hospitals (Table 1). A total of 117 culture-proven EOS cases were found, resulting in a culture-proven EOS incidence of 1.1 (0.11%, 95% CI 0.09–0.14) cases per 1000 live births. The most common pathogens were *Streptococcus agalactiae* (40.1%), *Escherichia coli* (23.2%) and *Staphylococcus aureus* (13.3%). Table 2 presents all pathogens identified. Antibiotic therapy was initiated in the first 3 days of life in 4755 neonates and continued for more than 48 h in 2399 neonates. The overall proportion of neonates initiated on antibiotics was 4.6% (95% CI 4.5–4.7). In 50.5% (95% CI 49.0–51.9), therapy was continued for more than 48 h, leading to a total proportion of neonates being treated with antibiotics for more than 48 h of 2.3% (95% CI 2.2–2.4). For each case of proven EOS, 40.6 neonates were started on antibiotic therapy, and 20.5 neonates were continued on antibiotics for more than 48 h. We found a moderate correlation between EOS incidence and antibiotic initiation across hospitals (Spearman r = 0.62; 95% CI, 0.14–0.86). No significant correlation was found between EOS incidence and therapy continuation.

### 2.2. Variation among Hospitals

Regional Hospitals (Hospitals A–M):

The culture-proven EOS incidence among regional hospitals was 0.9 (0.09%, 95% CI 0.08–0.12) cases per 1000 live births. The overall proportion of neonates started and continued on antibiotics were 3.8% (95% CI 3.7–3.9) and 2.2% (95% CI 2.2–2.3), respectively. Figure 2 shows per hospital the number of treated neonates per culture-proven EOS case. The number of neonates initiated on antibiotics per culture-proven EOS case ranged from 11 to 90. The number of neonates continued on antibiotics for more than 48 h per culture-proven EOS case ranged from 8 to 56. Details per individual hospital per year can be found in Appendix A.

Academic Hospitals (Hospital N and O):

The culture-proven EOS incidence in academic hospitals was 2.7 (0.27%, 95% CI 0.19–0.39) cases per 1000 live births. The overall proportion of neonates started and continued on antibiotic therapy in academic hospitals were 11.4% (95% CI 10.8–12.4) and 3.0% (95% CI 2.7–3.3), respectively. Figure 2 shows per hospital the number of treated neonates per culture-proven EOS case. The number of neonates initiated on antibiotics per culture-proven EOS case, ranged from 32 to 79. The number of neonates continued on antibiotics for more than 48 h per culture-proven EOS case ranged from 4 to 36. Details per individual hospital per year can be found in Appendix A.

### 2.3. Academic vs. Regional Hospitals

Table 2 displays the relative risks of the academic versus the regional hospital population. The relative risk of culture-proven EOS was significantly higher (RR 2.93, 95% CI, 1.93–4.42) in the academic hospital population compared to the regional hospital population. The relative risk of antibiotic initiation was also significantly higher (RR 2.99, 95% CI 2.81–3.18) in the academic hospitals, compared to the regional hospitals, yet similar numbers of neonates were initiated on antibiotics per culture-proven EOS case (RR 1.00, 95% CI 0.99–1.01). The relative risk of treatment continuation for more than 48 h in treated neonates was significantly lower in the academic hospital population (RR 0.44, 95% CI 0.40–0.48) with a remarkable difference between the two academic hospitals. The relative risk of treatment continuation for more than 48 h in all live-born neonates was significantly higher (RR 1.32, 95% CI 1.18–1.48) in the academic hospital population, although the relative risk of antibiotic therapy continuation per culture-proven EOS case was significantly lower (RR 0.94, 95% CI 0.91–0.98). 

### 2.4. Differences within a Three-Year Period

No significant differences in initiation and continuation of antibiotic therapy were observed between the years 2019, 2020 and 2021. In Appendix A, initiation, continuation and incidence rates can be found per hospital per year. 

## 3. Discussion

This study aimed to determine antibiotic initiation and continuation rates for suspected EOS, along with the incidence of culture-proven EOS in The Netherlands. We found that across the hospitals in this study, including 13 regional and 2 academic hospitals, for each case of culture-proven EOS, 40.6 neonates were started on antibiotic therapy, and in 20.5 neonates treatment was continued for more than 48 h. The overall incidence of culture-proven EOS was 1.1 cases per 1000 live births. Of all neonates born in the hospital, 4.6% was started on antibiotic therapy in the first 3 days of life. Antibiotic treatment was continued for more than 48 h in 50.5% of these neonates, leading to a percentage of 2.3% of all neonates receiving antibiotic therapy for more than 48 h. We found large variations between hospitals in the numbers of neonates initiated and continued on antibiotics per culture-proven EOS case. Overall, no trends in time were seen for therapy initiation and continuation during the three-year period. 

The observed incidence of 0.9 culture-proven EOS cases per 1000 live births in the regional hospital population in our study falls within the range of reported EOS incidences among late preterm and term neonates in other high-income countries, varying between 0.13 and 1.45 per 1000 live births [20,24,25,26,27,28,29,30,31,32]. Predominant pathogens identified were *Streptococcus agalactiae*, *Escherichia coli* and *Staphylococcus aureus*, similar to the pathogen distribution in other Western countries [19,28]. The proportion of 3.8% of neonates that were started on empiric antibiotics in regional hospitals aligns with earlier reported treatment rates in late preterm and term neonates in high-income countries, which vary between 1.2% and 14.0% [12]. While the rate found in this study leans towards the lower end of this spectrum, it is noteworthy that, compared to other European countries such as Sweden, Norway, and Switzerland, all reporting antibiotic initiation rates of less than 3%, The Netherlands demonstrates a relatively higher percentage of neonates initiated on antibiotic therapy [13,19,20,21]. Comparing results regarding treatment continuation proves challenging due to variations in study objectives across existing literature. In our study, we found that 50.5% of neonates were still receiving antibiotics after 48 h. Recent studies in The Netherlands showed that despite negative blood cultures, antibiotic treatment was continued for more than 72 h in 31.5% and 35.9% of neonates [22,29]. Other studies have reported median treatment durations of 3–7 days in neonates without culture-proven EOS [19,30,31]. Despite differing study objectives, the common finding is a disproportionately high number of neonates receiving antibiotics after the blood culture remains negative. 

In the academic hospital population, a significant higher EOS incidence of 2.7 cases per 1000 live births was observed, which might be explained by the higher risk population in academic hospitals, as EOS incidence has shown to be higher in neonates born at lower gestational age [23,24,25,26,27]. Interestingly, though treatment initiation rates were significantly higher in the academic hospital population, a significantly lower percentage of treated neonates continued antibiotic therapy beyond 48 h, resulting in a lower treatment burden per confirmed EOS case compared to regional hospital population. This was an unforeseen finding, as we expected to see higher treatment continuation rates due to the generally sicker and more preterm population in academic hospitals. However, a possible explanation for this outcome could be the greater emphasis on antibiotic stewardship in academic hospitals, leading to more critical consideration of continuing antibiotic treatment. It is noteworthy, however, that this decreased continuation rate is solely attributable to one of the two academic hospitals. Specifically, in academic hospital N, 36 neonates were continued on antibiotics, whereas in academic hospital O, 4 neonates were continued on antibiotics for each case of culture-proven sepsis. 

The clear variations across hospitals in the number of neonates initiated and continued on antibiotic therapy for each case of culture-proven EOS are consistent with prior research showing large variations in treatment rates between hospitals in high-income countries [18,19]. As nearly all hospitals in our study reported to use the same Dutch guideline during 2019–2021, namely, “Prevention and Treatment of Early-onset Neonatal Infections (2017)”, it is particularly interesting to see such large variations in clinical practice [32]. This raises questions about the underlying factors influencing policy choices in daily practice. However, the identified discrepancies between hospitals resonate with a recent study that reported low adherence to the current national Dutch guideline [22]. 

### 3.1. Strengths/Limitations

With our cohort, we managed to include approximately 25% of all live-born neonates in The Netherlands. As we covered a three-year period, we were able to capture possible time-related differences in care. Participating hospitals were both academic and regional hospitals and covered all different regions, resulting in a representative sample of Dutch neonatal care. Our study also has several limitations. First, to obtain a high rate of included hospitals we aimed to decrease the burden of data collection for the participating hospitals by keeping the number of different indicators to a minimum. We did not collect information on reasoning for the start or continuation of antibiotic therapy, mortality and total duration of antibiotic therapy. Especially data on the exact duration of antibiotic therapy would have been useful to compare results to other studies. We chose the 48-h threshold based on the Dutch national guideline, which recommends to stop antibiotic therapy after 36–48 h in case of a negative blood culture and both the CRP-levels and clinical condition of the child are reassuring. Our study lacks data on therapy discontinuation beyond 48 h, which could be particularly pertinent for the 48- to 72-h timeframe, where logistical factors such as shift changes, laboratory operating hours or medication stop orders might contribute to delays. Second, our calculation of birth numbers relied on in-hospital births, thereby excluding home births, potentially resulting in an overestimation of EOS incidence and treatment rates. Approximately 17% of neonates are born at home in The Netherlands annually, yet specific regional data are unavailable, preventing us from implementing a proper correction. However, since home-born neonates are also treated in hospital for EOS, the ratio between the number of treated neonates and the number of neonates with proven sepsis remains unchanged. Third, since participation in our study was voluntary, participation bias might be present. Hospitals that are already concerned with antibiotic overuse and working toward lowering the antibiotic exposure might be more prone to participate in our study, potentially leading to an underestimation of antibiotic use in The Netherlands. However, considering the large variation in clinical management between participating hospitals in this study with both low and high treatment rates, we expect the influence of participation bias on our results to be minimal. Lastly, data collection was conducted by the participating hospitals themselves. While we provided clear instructions and offered feedback on data collection, we lack insight into the specific methodologies employed by each hospital for data generation. 

### 3.2. Clinical Implications

This research is a necessary first step to improve antibiotic stewardship as it provides clear insight for physicians into the high burden of antibiotic use for EOS among the Dutch neonatal population. Giannoni. et al. suggested to pursue a neonatal antibiotic treatment rate of less than 1%, which was also the lowest initiation rate we found in our study, concomitant with a continuation rate of 0.5% [19]. In our view, these rates are both realistic and progressive to pursue. The next step is creating a standardised open access dashboard where key indicators on treatment rates versus EOS incidence are transparently reported [16]. Simultaneously, implementation of a revised guideline including strategies that are proven to reduce overtreatment, such as the EOS calculator or serial clinical examinations, should be pursued [33,34,35,36]. Moreover, a key area of improvement lies in the common decision of physicians to continue treatment despite a negative blood culture [29,37]. Though blood cultures are the golden standard for diagnosing EOS, physicians report multiple reasons for treatment continuation despite a negative culture. Reported reasons are concerns about incorrectly obtained cultures, low bacterial blood concentrations, intrapartum antibiotics, elevated CRP levels and clinical symptoms possibly reflecting sepsis [29,37]. However, as blood cultures with a minimal volume of 1 mL obtained before antibiotic initiation have high sensitivity, also for detecting low bacterial concentrations, there should be an emphasis on improving physicians’ confidence in blood cultures [38,39]. Moreover, using biomarkers for EOS management should be brought to a minimum, especially in the early stage when starting antibiotics, and only be applied when used correctly. Biomarkers alone cannot be used to determine the presence of infection, as they have a low positive predictive value. Some studies have demonstrated that EOS management without routine use of biomarkers significantly reduces antibiotic use [40,41]. However, in cases of substantial uncertainty, biomarkers may help physicians deciding to stop therapy [42,43]. Incorporating these strategies into clinical practice, combined with insight in actual antibiotic prescriptions and EOS incidence, may give an impulse to appropriate antibiotic use.

## 4. Methods

### 4.1. Study Design, Population and Period

We conducted a retrospective study in The Netherlands during a three-year period between 1 January 2019 and 31 December 2021, exploring antibiotic treatment exposure for suspected EOS among preterm and term neonates. In The Netherlands, neonatal care is distributed across 70 hospitals, including 7 academic hospitals and 63 regional hospitals All academic hospitals are equipped with Level IV facilities, providing the highest level of care to neonates born at or after 24 weeks of gestation. Regional hospitals have Level I or II facilities and offer care only for infants born at or after 32 weeks of gestation [44]. Mothers expected to deliver before 32 weeks are typically referred to academic hospitals, making the birth of neonates under 32 weeks in regional hospitals a rare occurrence. All 70 hospitals were approached by the research team to participate in this study via a standardised email followed by two reminder emails. All live-born neonates born in one of the participating hospitals during the study period were eligible for inclusion. No exclusion criteria applied.

### 4.2. Data Collection

For each participating hospital, the following key variables were recorded per calendar year: (1) total number of live births, (2) total number of neonates started on intravenous antibiotic therapy in the first 3 days of life for suspected EOS, (3) total number of neonates started on intravenous antibiotic therapy in the first 3 days of life for suspected EOS, treated for more than 48 h, (4) the number of positive blood cultures and (5) determination and interpretation of positive blood cultures (EOS-related pathogen or contamination). To obtain data, two different methods were employed depending on local facilities. The first method was consultation of the central database of the hospital’s business intelligence unit, where search terms were applied to find the required data. In case this was not available, obstetrics, microbiology and pharmacy databases were consulted separately by the local data manager. Information on local agreements on minimal gestational age at birth as well as the guideline used for management of early-onset sepsis during 2019–2021 were requested from the local paediatricians. Percentages of home births per year were obtained from the Dutch National Registry (LNR Perined) [45]. 

### 4.3. Study Outcomes and Definitions 

The primary outcomes of this study were the overall proportion of neonates started on intravenous antibiotic therapy within the first 3 days of life for suspected EOS (initiation), the overall proportions of neonates receiving antibiotic therapy for more than 48 h (continuation), and the incidence of newly developed culture-proven EOS including pathogen determination. Second, variation across hospitals in antibiotic initiation and continuation rates per culture-proven EOS case were investigated. Third, antibiotic initiation and continuation rates and EOS incidence were compared between regional and academic hospitals. Fourth, data were analysed to investigate whether there were differences in antibiotic initiation and continuation rates across the three years of the study period. Suspected EOS was defined as the presence of at least one maternal risk factor or neonatal clinical symptom described in the national guideline, prompting the clinician to initiate a course of empirical antibiotic treatment within the first three days of life [34]. Culture-proven EOS was defined as blood culture-proven infection in the first three days of life with an EOS-related pathogen, as interpreted by the medical microbiology department and/or paediatrician of the participating hospital. For blood cultures, blood was sampled into a single aerobic paediatric bottle and placed in a continuously monitoring microbial detection system. If a blood culture tested positive, a Gram stain of the positive bottle was performed and subcultures were made to further characterise the bacteria. A blood culture was considered negative if no bacterial growth was observed within 36–48 h after collection, with the exact cut-off determined by local hospital guidelines

### 4.4. Statistical Analysis

Categorical data were presented as frequencies (number and percentages) with 95% confidence intervals (CI) for proportions. Continuous data were presented as means with standard deviations (SD) when data are normally distributed, and as median with interquartile ranges (IQR) when data were non-normally distributed. Spearman’s rank correlation was used to explore correlation between treatment rates and incidence of EOS. Relative risks with 95% confidence intervals were calculated for comparison of academic hospitals (Level IV facilities) with regional hospitals (Level I–II facilities). To compare initiation and continuation proportions per year, Chi-square tests of independence were performed. *p*-Values < 0.05 are considered significant. Analyses were performed using IBM SPSS Statistics 29.0.

### 4.5. Ethical Considerations

The study was approved by the Medical Ethical Review Committee of the Tergooi MC under number 22.036 and was not subjected to the Medical Research Involving Human Subject Act. The data were anonymised using Castor™ Electronic Data Management System and our study complied with the General Data Protection Regulation. A Data Transfer Agreement was signed between participating hospitals and the study team.

## 5. Conclusions

Early postnatal antibiotic use in The Netherlands is high compared to EOS incidence. Moreover, significant variations are observed in the antibiotic treatment rates across hospitals despite the same applicable guideline, indicating the use of diverse other strategies in daily clinical practice. Future research should focus on the content of these strategies and study their association with the level of antibiotic treatment. The observed variations show both the urgent need and the potential for implementing revised guidelines, including current available interventions that help reduce antibiotic exposure. Data on key indicators of EOS management should be transparently collected and shared to enhance current antibiotic stewardship practices and support research on the most effective EOS management strategies.

## Figures and Tables

**Figure 1 antibiotics-13-00537-f001:**
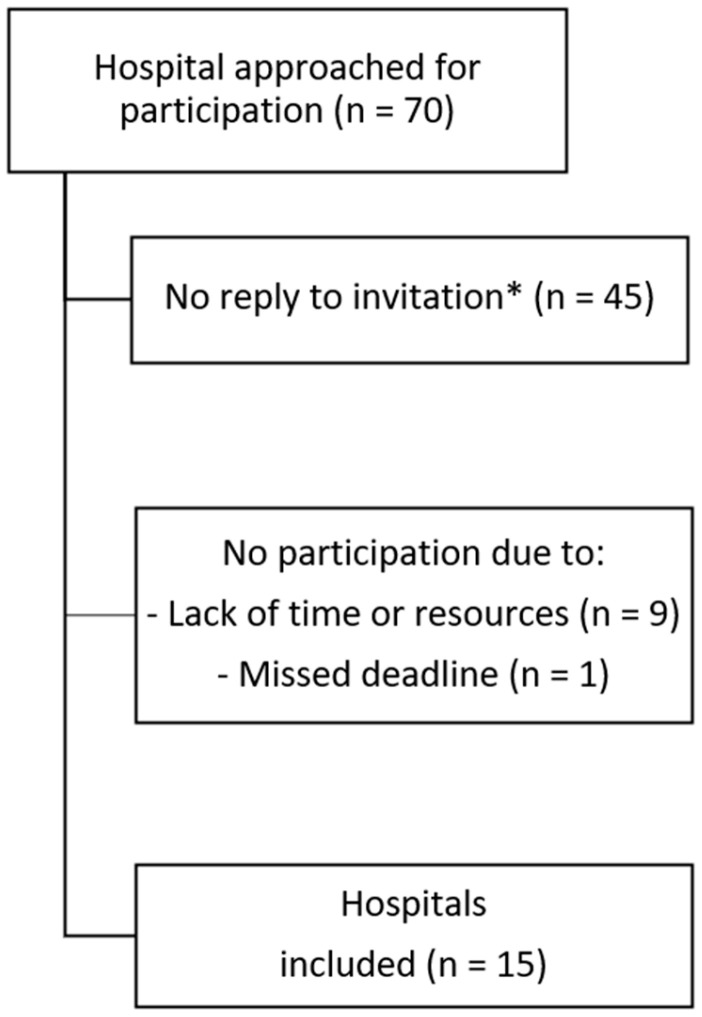
Flowchart of hospital inclusions. * The reason for the lack of response was not formally requested and is therefore not known.

**Figure 2 antibiotics-13-00537-f002:**
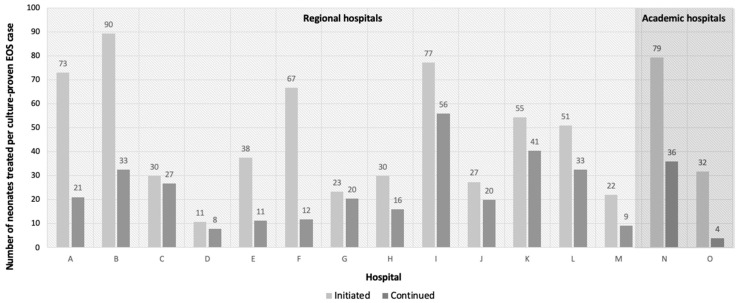
Number of neonates initiated and continued >48 h on antibiotic therapy per culture-proven EOS case.

**Table 1 antibiotics-13-00537-t001:** Main outcomes for the overall cohort, academic hospital population and regional hospital population.

	All Hospitals N = 15 n (%, 95% CI)	Academic Hospitals (Level IV Facilities) N = 2 n (%, 95% CI)	Regional Hospitals (Level I–II Facilities) N = 13 n (%, 95% CI)	Academic vs. Regional
				Relative Risk (95% CI) ^e^
Number of births	103492	10912	92580	NA
Culture-proven EOS ^a^	117 (0.11%, 0.09–0.14)	30 (0.27%, 0.19–0.39)	87 (0.09%, 0.08–0.12)	2.93 (1.93–4.42)
Antibiotic therapy initiation ^b^	4755 (4.6%, 4.5–4.7)	1240 (11.4%, 10.8–12.0)	3515 (3.8%, 3.7–3.9)	2.99 (2.81–3.18)
Antibiotic therapy continuation >48 h/total number of neonates initiated	2399 (50.5%, 49.0–51.9)	323 (26.0%, 23.6–28.6)	2076 (59.1%, 57.4–60.7)	0.44 (0.40–0.48)
Antibiotic therapy continuation >48 h/total number of births	2399 (2.3%, 2.2–2.4)	323 (3.0%, 2.7–3.3)	2076 (2.2%, 2.2–2.3)	1.32 (1.18–1.48)
Number initiated per culture-proven sepsis case ^c^	40.6	41.3	40.4	1.00 (0.99–1.01)
Number continued per culture-proven sepsis case ^d^	20.5	10.8	23.9	0.94 (0.91–0.98)

^a^ Incidence of positive blood cultures growing pathogenic bacteria as defined by the local microbiologist/paediatrician; ^b^ Initiation of antibiotic therapy for suspected early-onset sepsis (EOS) within the first 3 days after birth; ^c^ Number of neonates initiated on antibiotic therapy for each case of culture-proven sepsis; ^d^ Number of neonates continued on antibiotic therapy after 48 h for each case of culture-proven sepsis; ^e^ Relative risk of academic hospitals (compared to regional hospitals) with 95% confidence interval.

**Table 2 antibiotics-13-00537-t002:** Determination of positive blood cultures that were defined as pathogenic by the local microbiologist/paediatricians.

Pathogen Determination	All Positive Blood Cultures	Academic Hospitals	Regional Hospitals
	N = 117	N = 30	N = 87
*Streptococcus agalactiae*	47 (40.1%)	9 (30.0%)	38 (43.7%)
*Escherichia coli*	25 (23.3%)	7 (23.3%)	18 (20.7%)
*Staphylococcus aureus*	13 (11.1%)	5 (16.7%)	8 (9.2%)
*Streptococcus mitis*	5 (4.3%)	0 (0.0%)	5 (5.7%)
*Streptococcus anginosus*	4 (3.4%)	2 (6.7%)	2 (2.3%)
*Enterococcus faecalis*	4 (3.4%)	1 (3.3%)	3 (3.4%)
*Staphylococcus epidermidis* *	3 (2.6%)	1 (3.3%)	2 (2.3%)
*Listeria monocytogenes*	3 (2.6%)	3 (10.0%)	0 (0.0%)
*Other pathogens* **	13 (11.1%)	3 (10.0%)	10 (11.5%)

* These pathogens belong to the coagulase negative staphylococcus species (CoNS); ** Among ‘other pathogens’, the following pathogens were identified: *Streptococcus gallolyticus* (2), *Staphylococcus capitis* * (1), *Streptococcus pneumoniae* (1), *Streptococcus pyogenes* (2), *Staphylococcus haemolyticus* * (2), *Streptococcus parasanguinis* (1), *Klebsiella oxytoca* (1), *Sphingomonas paucimobilis* (1) *Haemophilus influenzae* (2).

## Data Availability

Deidentified data will be shared upon request.

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
