# Peer review of "Incidence of Antibiotic Exposure for Suspected and Proven Neonatal Early-Onset Sepsis between 2019 and 2021: A Retrospective, Multicentre Study"

_antibiotics, 2024, doi:10.3390/antibiotics13060537_

Round 1

Reviewer 1 Report

Comments and Suggestions for Authors

In general, the article is well written, understandable and accessible. The topic covered is interesting and important from the point of view of public health (antibiotics) and the health of newborns.

 In the first paragraph of the article (89 words), the authors used 13 bibliographic references, with only one from 2023, while all the others are from earlier years. It is worth reconsidering if this makes sense, considering the available information and the relevance of the articles.

  The idea that begins on line 74 'Relatively recent data...' and ends on line 81 lacks bibliographic support

 To our knowledge, there has been no recent extensive assessment of antibiotic usage 83

for suspected EOS in Dutch neonates” – improve this sentence by giving it a less personal meaning

 Introduce the methodology paragraph following the Introduction (instead of it being point 4).

 Considering Figure 1, it might be worthwhile to analyze or reflect on the Health Units that chose not to respond to the request. This could be interesting to understand the possible reasons

 Perhaps analyze the most frequently identified microorganisms by group (academic and regional hospital)

 “In our study” –Perhaps analyze the most frequently identified microorganisms in this study by group

 In our opinion, the combination

of a high burden of antibiotic treatment and low guideline adherence on one hand, and increasing awareness of both short- and long-term side effects of early antibiotics on the other hand shows the urgent need for guideline revision and implementation of current  available interventions to reduce antibiotic exposure in neonates. – it will be a sentence more adapted to the conclusion than to the discussion

Develop the conclusion a little further, with ideas to implement and solve the problem encountered.

Comments on the Quality of English Language

The writing is correct, naturally some details could be improved

Author Response

Dear reviewer, 

Thank you for your time and attentive reading of our manuscript. We greatly value your comments and have incorporated them to the best of our ability. If a particular point could not be processed, we have provided explanations for this. For each point, our responses and adjustments to the manuscript are described in the attached file. 

Reviewer 2 Report

Comments and Suggestions for Authors

Dear authors, congratulations on your effort to assess the adequate use of antibiotic therapy in early-onset neonatal sepsis (EOS). 

The abstract is commendably well-structured and effectively highlights the key findings of the study, providing a concise overview of your research.

The aim of the study, materials, and methods are well described and the results are well presented. However, I would suggest defining academic and regional hospitals in terms of competency in neonatal care. 

Discussions overview all the presented results.

The limitations of the study are well-underlined and explained.

The conclusions are correct.

Author Response

Dear reviewer, 

Thank you for your time and attentive reading of our manuscript. We greatly value your comment and have incorporated it to the best of our ability. Our response and adjustment to the manuscript are described in the attachment. 

Reviewer 3 Report

Comments and Suggestions for Authors

The authors have conducted a retrospective multicenter study to estimate the incidence of antibiotic exposure (ATB) in cases of early-onset sepsis (EOS) in the Netherlands. The work covers statistical data from 15 hospitals of different levels of complexity throughout the country. The authors obtained results that will serve as a baseline for future statistics. The conclusions highlight the importance of transparency in administrative notification and the importance of implementing systematic studies and clinical guidelines to guide doctors in the best treatment of neonates with suspected sepsis.

The attached file indicates the minor observations: unifying the objective throughout the manuscript, completing the table titles, and correcting the names of the bacteria.

Author Response

Dear reviewer, 

Thank you for your time and attentive reading of our manuscript. We greatly value your comments and have incorporated them to the best of our ability. For each point, our responses and adjustments to the manuscript are described in the attached document. 

Reviewer 4 Report

Comments and Suggestions for Authors

1. I would like to express my gratitude for the opportunity to review the manuscript entitled Incidence of antibiotic exposure for suspected and proven neonatal early-onset sepsis between 2019 and 2021: a retrospective, multicentre study” for possible publication in Antibiotics. I would like to praise the well-designed experiment, as well as the good analysis and presentation of results. However, there are some concerns that need to be addressed in order for the study to be considered for publication.

2. Based on content of the manuscript, I would like to discuss the appropriate usage of the terms "incidence" and "prevalence" in the title of your study.

3. The data analysis primarily involved calculating frequencies with 95% confidence intervals for proportions and odds ratios. While this provides valuable information, I would like to recommend considering the analysis of relative risk and incidence risk ratio.

4. To enhance the comprehensiveness of your study, I recommend including the inclusion criteria for early-onset neonatal sepsis in the manuscript. Please note that the specific details of the clinical inclusion criteria should be based on the guidelines followed by the participating hospitals.

5. Please provide additional information on blood cultures and bacterial identification methods

6.   I have a suggestion to rename Table 1.

7. I noticed a lack of written description regarding the causative pathogen test results in the article. Kindly consider expanding the discussion section to include a detailed analysis of the causative pathogen test results also.

8. If the author has already conducted antimicrobial susceptibility testing, incorporating this information into your manuscript would be highly valuable to the readers.

9. The article presented the results from regional and academic hospitals separately, but the authors did not clearly state their assumptions, hypothesis or objectives in the introduction. Furthermore, there is a lack of comparative criticism regarding this issue.

10.   The conclusion section should become more focused, informative, and impactful. It will effectively summarize the key findings, provide valuable insights, and offer suggestions for future research directions.

Author Response

Thank you for your time and attentive reading of our manuscript. We greatly value your comments and have incorporated them to the best of our ability. If a particular point could not be processed, we have provided explanations for this. For each point, our responses and adjustments to the manuscript are described in the attached document. 

Round 2

Reviewer 4 Report

Comments and Suggestions for Authors

After carefully reviewing your responses, I am delighted to inform you that I wholeheartedly accept the manuscript in its current form.